# Association between Cholesterol Level and the Risk of Hematologic Malignancy According to Menopausal Status: A Korean Nationwide Cohort Study

**DOI:** 10.3390/biomedicines10071617

**Published:** 2022-07-06

**Authors:** Wonyoung Jung, Keun Hye Jeon, Jihun Kang, Taewoong Choi, Kyungdo Han, Sang-Man Jin, Su-Min Jeong, Dong Wook Shin

**Affiliations:** 1Department of Family Medicine and Supportive Care Center, Samsung Medical Center, Sungkyunkwan University School of Medicine, Seoul 06351, Korea; wyjung.md@gmail.com; 2Department of Medicine, Sungkyunkwan University School of Medicine, Seoul 06351, Korea; 3Department of Family Medicine, CHA Gumi Medical Center, CHA University, Gumi 39295, Korea; kh6325209@naver.com; 4Department of Family Medicine, Kosin University Gospel Hospital, Kosin University College of Medicine, Busan 49267, Korea; josua85@naver.com; 5Division of Hematologic Malignancies and Cellular Therapy, Duke University Medical Center, Durham, NC 27710, USA; taewoong.choi@gmail.com; 6Department of Statistics and Actuarial Science, Soongsil University, Seoul 06978, Korea; hkd917@naver.com; 7Division of Endocrinology and Metabolism, Department of Medicine, Samsung Medical Center, Sungkyunkwan University School of Medicine, Seoul 06355, Korea; sjin772@gmail.com; 8Department of Clinical Research Design & Evaluation, Samsung Advanced Institute for Health Science & Technology (SAIHST), Sungkyunkwan University, Seoul 06355, Korea

**Keywords:** lipid, HDL-C, menopause, reproductive factor, hematologic malignancy

## Abstract

Recent studies have revealed the possible association between serum cholesterol levels and hematologic malignancy (HM). However, limited information is available about how reproductive factors interact with this association. Therefore, we investigated the roles of serum cholesterol in the risk of HM according to the menopausal status. We finally identified 1,189,806 premenopausal and 1,621,604 postmenopausal women who underwent a national health screening program in 2009 using data from the Korean National Health Insurance Service database. Overall, 5449 (0.19%) developed HM. Among postmenopausal women, the inverse associations were observed between total cholesterol, high-density lipoprotein cholesterol (HDL-C), low-density lipoprotein cholesterol (LDL-C) levels, and the risk of overall HM. In premenopausal women, the highest quartile of HDL-C was associated with a reduced risk of HM compared with the lowest quartile of HDL-C consistent with results in postmenopausal women (adjusted hazard ratio [aHR] 0.80, 95% confidence interval [CI] [0.68–0.95]), whereas the highest quartile of triglyceride (TG) showed an increased risk of HM compared to the lowest quartile of TG, (aHR 1.22, 95% CI [1.02,1.44]) only in premenopausal women. Our finding suggests that lipid profiles are differently associated with HM risk by menopausal status.

## 1. Introduction

Hematologic malignancy (HM), a cancerous condition in hematopoietic systems, refers to a group of heterogeneous diseases such as multiple myeloma (MM), non-Hodgkin’s lymphoma (NHL), Hodgkin’s lymphoma (HL), lymphoid leukemia (LL), and myeloid leukemia (ML). As their etiology is not fully understood, establishing the pathogenesis, and identifying the risk factors of HM are of clinical significance. 

Lipids play an important role in promoting cell growth and proliferation [1], as lipid metabolism disorders cause abnormal expression of various genes, proteins, and dysregulation of cytokines and signaling pathways [2]. Currently, the mechanisms underlying the association between serum lipid and HM are not clearly understood. According to the Alpha-Tocopherol Beta-Carotene Cancer Prevention study, elevated high-density lipoprotein cholesterol (HDL-C) levels are associated with a reduced risk of NHL, suggesting that anti-inflammatory and antioxidant properties of HDL-C may have protective role against cancers [3]. Lower low-density lipoprotein cholesterol (LDL-C) level might also increase the risk of HM by providing less coenzyme Q to the circulation, thus diminishing the body’s total cellular antioxidant capacity [4]. Another cohort study of the Austrian population observed that elevated triglyceride (TG) levels are associated with a reduced risk of NHL, although this association might result from cancer metabolism suggesting reverse causation [5]. A large retrospective study pooling seven European cohorts demonstrated an inverse association of total cholesterol and the risk of myeloid neoplasm in women, whereas no associations were found in TG [6], but lack of information on physical activity, alcohol consumption, and medication demands careful consideration in interpreting the results. Our prior research has demonstrated that low level of HDL-C was significantly associated with increased risk of HM, suggesting that a low HDL-C level is an independent risk factor and preclinical marker for HM [7]. Furthermore, high variability of HDL-C from repeated measurements was associated with an increased risk of developing MM [8].

Higher incidence of HM in male than female suggests potential role of sex hormone in the development of HM. However, limited information is available about how sex hormone and menopausal status affects the risk of HM [9,10,11]. Multicenter study using data from European Prospective Investigation into Cancer and Nutrition demonstrated no statistically significant associations between reproductive factors and NHL, including parity, age at first birth, breastfeeding, oral contraceptive use, or ever use of postmenopausal hormone therapy [9]. A case–control study of female adults with acute and chronic ML concluded that exogenous hormone uses, and reproductive factors are unlikely to have a significant role in the etiology of ML [10].

To the best of our knowledge, none of previous studies has examined the association between lipid profiles and risk of HM according to menopausal status, even though there is possibility that estrogen and lipid interact on the development of HM. During menopause transition, circulating estrogen level decreases, which might have detrimental effect on the lipid profiles [12,13,14]. 

Therefore, in this nationwide cohort study, we aimed to investigate the association between lipid levels and the risk of HM according to the menopausal status with assessing this association by risk of subtypes of HM.

## 2. Materials and Methods

### 2.1. Data Source and Study Setting

The National Health Insurance Service (NHIS) is a single insurer that provides mandatory universal coverage to 97% of the Korean population and administers a medical aid program to 3% of the population in the lowest income bracket, which is funded by general taxation. Medical service providers are reimbursed by NHIS for services provided. 

The NHIS also runs national health check-up programs, which include a cardiovascular health screening test for all those aged 40 and above and all employees regardless of age and provides reimbursement for screening services provided [15]. In addition, the NHIS also provides administrative oversight for the National Cancer Screening Program (NCSP), which currently includes screening for breast, and cervical cancers for all individuals, as indicated by age [16,17]. Therefore, the NHIS database comprises an eligibility database (age, sex, socioeconomic variables, type of eligibility, income level, etc.), a medical treatment database (based on medical bills that were claimed by medical service providers for their medical expense claims), and a health examination database (results of general health examinations and questionnaires on lifestyle and behavior). 

### 2.2. Study Population

Among female participants (age ≥ 30) who underwent general health screenings and NCSP for breast and/or cervical cancer in 2009, which requires to answer the self-questionnaire about reproductive factors, we excluded individuals with history of hysterectomy or missing information on menopausal status (*n* = 306,485), and those with pre-existing cancer diagnoses (*n* = 61,649). Participants who had a new HM diagnosis or died (*n* = 5564) within one year after the health screening date were also excluded to reduce possible reverse causality. Therefore, 2,811,410 participants were included in the final study population. The participants were followed from the day of health examination of 2009 to the occurrence of HM, death of any cause, or 31 December 2017, whichever came first.

### 2.3. Definition of Hematologic Malignancy

HM was identified using the following diagnoses from the International Classification of Diseases 10th revision (ICD-10): MM [C90.0], HL [C81], NHL [Diffuse Large B-cell lymphoma (C83.3), follicular lymphoma (C82)], lymphoid leukemia (LL) [chronic lymphocytic leukemia (C91.1) and acute lymphocytic leukemia (C91.0)], and ML [chronic myeloid leukemia (C92.1) and acute myeloid leukemia (C92.0, C92.5, C92.4, C92.6, C93.0, C94.0, C94.2)]. In addition to ICD-10 codes, we confirmed cases of HMs through the registration program for rare incurable diseases. Since 2009, the Korean government has provided co-payment reduction for registered cancer patients. However, only patients whose cancer diagnoses were confirmed by physicians (after thorough evaluation) could be registered in this program.

### 2.4. Measurement of Cholesterol Levels

Blood samples were collected after at least 8 hours of fasting on the day of the health examination in 2009. The cholesterol concentrations were measured enzymatically in each clinic or hospital accordance with universal standard and proper quality control of laboratory tests [18]. The LDL-C level was calculated using the Friedewald Equation when the TG level was <400 mg/dL. If the TG level exceeded 400 mg/dL, the LDL-C level was measured with the direct assay. The participants were categorized by quartile of baseline of each lipid profile level (Q1—the lowest, Q2, Q3, and Q4—the highest). Medical institutions and laboratories providing health screenings must be certified by the NHIS and have regular quality checks.

### 2.5. Covariates

The health insurance premium, a proxy of income level was classified into quartiles. Body mass index (BMI) was calculated as weight in kilograms divided by height in meters squared. Waist circumference (WC) was measured at the midpoint between the lower margin of the ribs at the mid-axillary plane and the top of the iliac crest. The participants provided information on lifestyle behaviors using standardized questionnaires. Smoking status was categorized as none, ex-, and current smokers. Alcohol consumption was categorized as none, mild, and heavy (≥30 g of alcohol consumption per day). Regular physical activity was defined if they exercised strenuously ≥ one time/week for at least 20 min per session. Comorbidities (hypertension, diabetes mellitus, and dyslipidemia) were comprehensively identified based on the combination of diagnosis codes (ICD-10) with relevant prescribed medications for each disease and clinical information. Arterial hypertension was defined as any of the following: systolic blood pressure ≥140 mmHg; diastolic blood pressure ≥90 mmHg; or treatment with an antihypertensive medication that was linked to the arterial hypertension ICD-10 codes (I10–I13 and I15). Diabetes mellitus was defined as a blood glucose level ≥126 mg/dL or history of a hypoglycemic medication prescription that was linked to a diabetes mellitus ICD-10 code (E11–E14). Dyslipidemia was defined as total cholesterol ≥240 mg/dL or history of a lipid-lowering medication that was associated with an ICD-10 code (E78).

### 2.6. Statistical Analysis

The comparison of baseline characteristics according to HM was conducted using *t*-test for continuous variables or the chi-square test for categorical variables. Subgroup analysis was performed according to menopausal status. The incidence rates of HM were assessed as the incident cases divided by 100,000 person years. In order to estimate the risk of HM for each quartile of lipid profiles including total cholesterol, HDL-C, LDL-C, and TG, Cox regression hazard model was used to examine the hazard ratios of HM using the lowest quartile as the reference group. Multivariable analyses were adjusted for age in model 2 and for age, income, BMI, smoking, alcohol consumption, regular exercise, diabetes mellitus, and history of taking medication for dyslipidemia within a year in model 3. To evaluate the potential effect modification by statin use, P for interaction was calculated.

The statistical analyses were performed using SAS version 9.4 (SAS Institute Inc., Cary, NC, USA). *p*-values < 0.05 were considered statistically significant. 

### 2.7. Ethics Statement

This study was approved by the Institutional Review Board of Samsung Medical Center (SMC 2020-04-141). Anonymized and de-identified information was used for analyses; therefore, informed consent was not required. The database is open to all researchers whose study protocols are approved by the official review committee.

## 3. Results

### 3.1. Baseline Characteristics

In total, 5449 (0.19%) participants were newly diagnosed with HM during median period of follow-up 8.4 (8.1–8.6) years, including 1188 (0.04%) HM events among premenopausal group, and 4261 (0.15%) among postmenopausal group (Table 1). The mean age of pre-and postmenopausal women in this study were 43.9 and 61.4 years old, respectively. 

The mean total cholesterol, HDL-C, and LDL-C levels were 200.8 ± 42.8 mg/dL, 59.1 ± 36.5 mg/dL, and 121.4 ± 73.5 mg/dL, respectively (Appendix A). Total cholesterol and HDL-C levels were lower, but TG level was higher in participants with HM than participants without HM. This difference of lipid levels by incidence of HM was consistent in postmenopausal women, whereas no significant differences except for TG were found in premenopausal women (Table 1). Comorbidities such as hypertension, diabetes mellitus, and dyslipidemia were more prevalent in HM group. 

### 3.2. Serum Lipid Levels and Risk of Hematologic Malignancy According to Menopausal Status

The association between each serum lipid profile and the incidence of HM was investigated according to menopausal status after adjusting for covariates (Table 2). Cut-off values of quartile of each lipid profile are demonstrated in Appendix A. The inverse associations were observed between total cholesterol (Q4: adjusted hazard ratio [aHR] 0.73, 95% confidence interval [CI] [0.68–0.79]), HDL-C (Q4: aHR 0.72, 95% CI [0.67–0.78]), LDL-C (Q4: aHR 0.79, 95% CI [0.73–0.85]) levels, and the risk of overall HM (all P for trend <0.001). The serum TG levels were not significantly associated with risk of HM. These associations were consistent in postmenopausal women, but not in premenopausal women. In postmenopausal women, participants with the highest quartiles of HDL-C, and LDL-C showed the 30% and 26% decreased risk of overall HM, respectively. In addition, participants with the highest total cholesterol showed the lowest risk of overall HM (aHR 0.67, 95% CI [0.61–0.73]).

In premenopausal women, the highest HDL-C quartile group showed 20% decreased risk of HM compared to the lowest HDL-C quartile group (Q4: aHR 0.80, 95% CI [0.68–0.95]). The highest TG quartile group showed increased risk of HM compared to the lowest TG quartile, (Q4: aHR 1.22, 95% CI [1.02–1.44]). In contrast to the findings in postmenopausal women, total cholesterol and LDL-C levels were not associated with risk of HM in premenopausal women. Forest plots of adjusted hazard ratios for overall HM by quartiles of serum lipid levels are shown in Figure 1A (premenopausal women),B (postmenopausal women).

### 3.3. Risk of Subtypes of Hematologic Malignancy According to Serum Lipid Levels and Menopausal Status

Among the 5449 HM participants, 1654 cases for MM, 189 cases for HL, 1965 cases for NHL, 437 cases for LL, and 1501 cases for ML were identified. Subgroup analyses were performed for each five subtypes of HM. 

Among postmenopausal women, higher levels of serum HDL-C were associated with the lower risk of MM (Q4: aHR 0.57, 95% CI [0.49–0.67]), HL (Q4: aHR 0.46, 95% CI [0.26–0.81]), NHL (Q4: aHR 0.83, 95% CI [0.71–0.96]), and ML (Q4: aHR 0.80, 95% CI [0.67–0.95]) compared to the lowest group of each lipid profile (Q1). Similarly, the inverse associations between LDL-C levels and the risk of MM (Q4: aHR 0.71, 95% CI [0.62–0.83]), NHL (Q4: aHR 0.73, 95% CI [0.63–0.84]), and ML (Q4: aHR 0.77, 95% CI [0.65–0.91]) were also observed compared to Q1 group in postmenopausal women. 

However, these associations were not significant in premenopausal women except for MM. A higher level of total cholesterol, HDL-C and LDL-C among premenopausal women was associated with a lower risk of MM. Details are described in Table 3. 

### 3.4. Stratified Analyses

After stratified analyses according to statin uses, the inverse association between total cholesterol, HDL-C, LDL-C levels, and the risk of HM was observed in statin non-users among postmenopausal women, which was consistent with our main findings. (Table 4) Among statin users, there was no significant association between total cholesterol, HDL-C, and LDL-C and risk of HM.

## 4. Discussion

To our knowledge, this is the first study that investigated on the association between serum lipid levels and HM according to menopausal status. Although numerous prior studies have examined the association between endocrine disease, including obesity, metabolic syndrome, diabetes, or dyslipidemia, and risk of HMs [6,19,20,21,22,23], none of them did not consider the difference by menopausal status. We clearly showed that the lower level of total cholesterol, HDL-C, LDL-C levels, but not TG, was associated with the risk of overall HM in female population. The different associations were found according to menopausal status: lower levels of total cholesterol, HDL-C, and LDL-C were associated with an increased risk of HM in postmenopausal women and lower HDL-C and higher TG level was associated with an increased risk of HM in premenopausal women. 

Although the precise mechanisms under this association are uncertain, lower lipid levels, in particular HDL-C might suggest loss of protective role against cancer development or result from cancer cell metabolism [7]. Low HDL-C levels may act as a surrogate marker for overall systemic inflammation [3] that affect oncogenes leading to leukemogenesis [24]. HDL-C suppresses myeloid proliferation and leukocytosis by decreasing granulocyte-monocyte progenitors and proliferation of interleukin-3 in bone marrow cells [25,26]. In addition, reduced HDL-C levels are closely related to insulin resistance [27], which leads to consecutive activation of insulin-like growth factor-1 (IGF-1) signaling that exhibits mitogenic activity via the PI3K/Akt cascade and RAS/RAF/MAPK/ERK pathway [28,29,30]. Mitogenic activity of IGF-1 was also found in both myeloid and lymphoid leukemia cell lines [31]. Metabolic change after menopause such as insulin resistance accompanied with estrogen deficiency could accelerate development of HM by making more vulnerable environments to cancer cell metabolism. In postmenopausal women, loss of protective effects of estrogen results in changes to body fat distribution that cause abdominal obesity [32], which is associated with insulin resistance [33]. Insulin resistance can in turn alter systemic lipid metabolism [34,35], resulting in high levels of TG, and low levels of HDL-C. Moreover, decreased circulating estrogen level itself promotes liver TG accumulation and leads to hepatic insulin resistance [36]. Previous in vitro investigations have demonstrated that sex hormones represent important factors influencing the risk of HM. Interferon regulatory factor 4 (IRF4) is highly expressed in B cells, plasma cells [37], and T cells [38]. Since estrogen plays a role in the inhibition of nuclear factor kappa B (NF-κB) which regulates transcription of IRF4 [39], difference in the incidence of HMs by menopausal status could be explained.

In this study, the highest TG was associated with an increased risk of HM in premenopausal women, but not in postmenopausal women. Several possible explanations could be postulated for these discrepant results according to menopausal status. In premenopausal women, it is possible that a higher TG level indicates severe insulin resistance considering protective effects of estrogen in TG levels [12,40]. Even though estrogen receptor β agonists showed the strong growth-suppressing effects on lymphoma and leukemia cells in mice in vitro [41,42], a higher TG reflected by insulin resistance might mitigate this protective estrogen effect. In contrast, serum TG levels were not associated with risk of HM which is consistent with previous studies. Previous clinical studies suggested that low levels of total cholesterol, HDL-C, and LDL-C but not TG can be accompanied by HMs such as MM and chronic ML [4,43,44], which may reinforce the cell membrane hypothesis since TG is not involved in cell membrane synthesis. In addition, a prospective study exhibited that elevated total cholesterol levels, but not TG levels were associated with the reduced risk of AML in women [6].

Regarding subtypes of HM, both pre-and postmenopausal women had a similar trend in the association between serum lipid profiles and risk of MM showing lower risk of MM in a higher level of total cholesterol, HDL-C and LDL-C. Previous in vitro studies have revealed that estrogen receptors-α were expressed in MM cells and anti-estrogen agents (e.g., tamoxifen) inhibited the proliferation of MM cells inducing MM cell apoptosis, although estrogen did not significantly alter MM cell proliferation [45] suggesting that difference in estrogen levels according to menopause is less likely to change the association between lipid profiles and risk of MM. In terms of HL, an uncommon lymphoma, postmenopausal women showed that a higher total cholesterol was associated with a decreased risk of HL, whereas no significant association was found in premenopausal women. However, this could be a chance finding considering of low incidence of HL in both groups. MM and HL showed more prominent association with serum lipid profiles than other HMs. 

In our study, statin users with low levels of total cholesterol, and LDL-C were not related to a significantly increased risk of HM, whereas statin non-users with low levels of total cholesterol, and LDL-C were significantly related to an increased risk of HM. In this context, we assumed that underlying cause for lower cholesterol is more important factor rather than low cholesterol levels itself. Some biological mechanisms could be suggested. Previous transcriptional profiling study hypothesized that genes involved in cholesterol metabolism are upregulated in cancer cells [46]. Therefore, it can be postulated that it may be not the cholesterol level that directly links to the risk of HM, but the shared mechanism between cholesterol metabolism and the development of cancer cells. Further trials may be needed to confirm this association. In addition, systematic reviews demonstrated that statin has a potential preventive effect on the risk of HM [47,48], suggested by the hypothesis of anti-inflammation and immunomodulation [49,50].

There are several limitations in this study. First, study population is from the Korean National Health Insurance Service database, which may yield ethnic bias as there are significant ethnic disparities in the incidence of HM [51]. Therefore, the results cannot be generalizable to other ethnic groups. Second, since there are a few different options for dyes and analyzers in enzymatic method, it is not possible to rule out discrepancy in laboratory findings between clinics or hospitals. With universal standard and proper quality control of laboratory test, however, consistent findings with well-known clinical information were reported by previous studies using the Korean National Health Insurance service database including lipid profiles [52,53]. Third, unmeasured confounding factors which are related to HM incidence such as occupational exposures were not controlled. 

In conclusion, our population-based study shows that low HDL-C level was associated with increased risk of HM both in premenopausal and postmenopausal women, but high TG level was associated with increased risk of HM only in premenopausal women. Further investigation to clarify the biological mechanisms by which cholesterol metabolism according to reproductive factors contribute to the development of HM is needed.

## Figures and Tables

**Figure 1 biomedicines-10-01617-f001:**
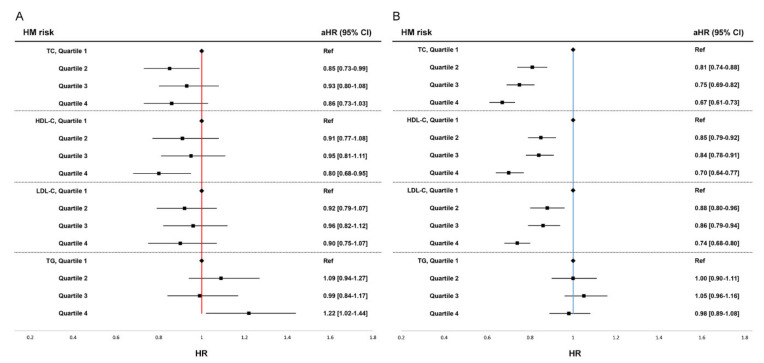
Forest plot of adjusted hazard ratio for overall hematologic malignancy (HM) by quartiles of serum lipid levels. (**A**) Risk of HM in premenopausal women, (**B**) Risk of HM in postmenopausal women. HR, hazard ratio; CI, confidence interval; Pre, premenopausal women; Post, postmenopausal women HRs are adjusted for age, body mass index, smoking, alcohol consumption, regular exercise, diabetes mellitus, and history of taking medication for dyslipidemia within a year.

**Table 1 biomedicines-10-01617-t001:** Baseline characteristics of the study population.

		Premenopause (*n* = 1,189,806)				Postmenopause (*n* = 1,621,604)			
		Total	Hematologic Malignancy		*p*-Value	Total	Hematologic Malignancy		*p*-Value
			No (*n* = 1,188,618)	Yes (*n* = 1188)			No (*n* = 1,617,343)	Yes (*n* = 4261)	
Age (years)		43.9 ± 5.5	43.9 ± 5.5	45.8 ± 5.8	<0.001	61.4 ± 8.7	61.4 ± 8.7	64.2 ± 8.2	<0.001
BMI (kg/m^2^)		23.1 ± 3.1	23.1 ± 3.1	23.6 ± 3.4	<0.001	24.2 ± 3.2	24.2 ± 3.2	24.4 ± 3.2	<0.001
WC (cm)	<75	624,274 (52.47)	623,727 (52.47)	547 (46.04)	<0.001	418,496 (25.81)	417,555 (25.82)	941 (22.08)	<0.001
	75–85	430,401 (36.18)	429,930 (36.18)	471 (39.65)		746,312 (46,02)	744,397 (46.03)	1915 (44.94)	
	≥85	135,131 (11.36)	134,961 (11.35)	170 (14.31)		456,796 (28.17)	455,391 (28.16)	1405 (32.97)	
TC *		190.8 ± 39.1	190.8 ± 39.1	191.5 ± 35.1	0.489	208.1 ± 44.0	208.1 ± 43.9	201.8 ± 39.2	<0.001
HDL-C *		60.6 ± 35.0	60.6 ± 34.9	60.4 ± 43.1	0.839	58.0 ± 37.6	58.0 ± 37.6	56.1 ± 40.6	0.001
LDL-C *		113.5 ± 75.5	113.5 ± 75.5	114.1 ± 55.3	0.770	127.2 ± 71.5	127.2 ± 71.5	124.3 ± 76.6	0.007
TG ^a^,*		87.4 (87.3–87.4)	87.4 (87.3–87.4)	93.7 (90.9–96.6)	<0.001	116.8 (116.7–116.8)	116.8 (116.7–116.8)	119.1 (117.3–120.9)	<0.001
Smoking status					0.457				0.041
Never		1,122,927 (94.4)	1,121,796 (94.4)	1131 (95.2)		1,558,971 (96.1)	1,554,846 (96.1)	4125 (96.8)	
Ex-smoker		23,284 (2.0)	23,265 (2.0)	19 (1.6)		18,243 (1.1)	18,197 (1.1)	46 (1.1)	
Current smoker		43,595 (3.7)	43,557 (3.7)	38 (3.2)		44,390 (2.7)	44,300 (2.7)	90 (2.1)	
Alcohol consumption					0.082				<0.001
Non		836,730 (70.3)	835,881 (70.3)	84 9(71.5)		1,415,969 (87.3)	1,412,099 (87.3)	3870 (90.8)	
Mild (<30 mg/d)		338,891 (28.5)	338,558 (28.5)	333 (28.0)		197,027 (12.2)	196,648 (12.2)	379 (8.9)	
Heavy (≥30 mg/d)		14,185 (1.2)	14,179 (1.2)	6 (0.5)		8608 (0.5)	8596 (0.5)	12 (0.3)	
Regular exercise		194,316 (16.3)	194,114 (16.3)	202 (17.0)	0.531	292,853 (18.1)	292,102 (18.1)	751 (17.6)	0.460
Systolic BP **		116.6 ± 14.2	116.6 ± 14.2	117.9 ± 14.3	0.003	125.7 ± 16.2	125.7 ± 16.2	126.7 ± 16.1	<0.001
Diastolic BP **		72.8 ± 9.9	72.8 ± 9.9	73.2 ± 9.9	0.189	76.9 ± 10.2	76.9 ± 10.2	77.2 ± 10.1	0.160
Fasting glucose *		93.3 ± 17.6	93.3 ± 17.6	94.4 ± 20.9	0.030	99.7 ± 24.5	99.7 ± 24.5	101.0 ± 25.1	0.001
Hypertension, yes		158,000 (13.3)	157,784 (13.3)	216 (18.2)	<0.001	750,286 (46.3)	748,035 (46.3)	2251 (52.8)	<0.001
Diabetes mellitus, yes		40,322 (3.4)	40,264 (3.4)	58 (4.9)	0.004	213,940 (13.2)	213,246 (13.2)	694 (16.3)	<0.001
Dyslipidemia, yes		126,916 (10.7)	126,775 (10.7)	141 (11.9)	0.179	554,355 (34.2)	552,972 (34.2)	1383 (32.5)	0.017

Data are expressed as mean ± standard deviation or number (%), except for triglycerides, which are presented as media (interquartile range) using the Wilcoxon rank-sum test. WC, Waist circumference; BP, blood pressure; eGFR, estimated glomerular filtration rate; TC, Total cholesterol; HDL-C, high-density lipoprotein cholesterol; LDL-C, low-density lipoprotein cholesterol; TG, triglycerides. * Unit: mg/dL, ** Unit: mmHg. ^a^ Geometric mean (95% confidence interval).

**Table 2 biomedicines-10-01617-t002:** Risk of hematologic malignancy by quartile of each lipid profile according to menopausal status.

		Total (*n* = 2,811,410)				Premenopause (*n* = 1,189,806)				Postmenopause (*n* = 1,621,604)			
Lipid Profile		Subjects (N)	Event (*n*)	IR	Model 3 aHR (95% CI)	Subjects (N)	Event (*n*)	IR	Model 3 aHR (95% CI)	Subjects (N)	Event (*n*)	IR	Model 3 aHR (95% CI)
TC	Q1 (low)	710,529	1442	246.0	1 (Ref.)	399,781	396	119.2	1 (Ref.)	310,748	1046	411.9	1 (Ref.)
	Q2	692,969	1277	222.7	0.83 (0.77–0.90)	332,944	304	109.8	0.85 (0.73–0.99)	360,025	973	328.1	0.81 (0.74–0.88)
	Q3	700,027	1367	235.9	0.81 (0.75–0.88)	269,814	287	127.9	0.93 (0.80–1.08)	430,213	1080	304.2	0.75 (0.69–0.82)
	Q4 (high)	707,885	1363	232.9	0.73 (0.68–0.79)	187,267	201	129.2	0.86 (0.73–1.03)	520,618	1162	270.4	0.67 (0.61–0.73)
	P trend				<0.001				0.145				<0.001
HDL-C	Q1 (low)	712,128	1798	306.3	1 (Ref.)	241,750	281	139.6	1 (Ref.)	470,378	1517	393.3	1 (Ref.)
	Q2	679,848	1332	236.8	0.86 (0.80–0.93)	277,350	282	122.2	0.91 (0.77–1.08)	402,498	1050	316.5	0.85 (0.79–0.92)
	Q3	733,666	1340	220.7	0.86 (0.80–0.93)	333,451	342	123.5	0.95 (0.81–1.11)	400,215	998	302.2	0.84 (0.78–0.91)
	Q4 (high)	685,768	979	172.5	0.72 (0.67–0.78)	337,255	283	101.1	0.80 (0.68–0.95)	348,513	696	242.0	0.70 (0.64–0.77)
	P trend				<0.001				0.021				<0.001
LDL-C	Q1 (low)	701,522	1380	238.6	1 (Ref.)	373,625	359	115.7	1 (Ref.)	327,897	1,021	380.9	1 (Ref.)
	Q2	689,154	1271	223.0	0.90 (0.84–0.97)	337,040	320	114.2	0.92 (0.79–1.07)	352,114	951	328.2	0.88 (0.80–0.96)
	Q3	712,207	1425	241.7	0.90 (0.84–0.97)	284,577	300	126.8	0.96 (0.82–1.12)	427,630	1,125	318.8	0.86 (0.79–0.94)
	Q4 (high)	708,527	1373	234.2	0.79 (0.73–0.85)	194,564	209	129.2	0.90 (0.75–1.07)	513,963	1,164	274.2	0.74 (0.68–0.80)
	P trend				<0.001				0.296				<0.001
TG	Q1 (low)	713,559	1014	171.2	1 (Ref.)	438,153	378	103.8	1 (Ref.)	275,406	636	278.9	1 (Ref.)
	Q2	687,369	1242	218.3	1.04 (0.96–1.13)	322,417	328	122.4	1.09 (0.94–1.27)	364,952	914	303.7	1.00 (0.90–1.11)
	Q3	703,944	1526	262.5	1.07 (0.99–1.16)	247,407	245	119.1	0.99 (0.84–1.17)	456,537	1281	341.1	1.05 (0.96–1.16)
	Q4 (high)	706,538	1667	286.5	1.04 (0.96–1.13)	181,829	237	156.8	1.22 (1.02–1.44)	524,709	1430	332.0	0.98 (0.89–1.08)
	P trend				0.352				0.100				0.794

Model 3 was adjusted for age, body mass index, smoking, alcohol consumption, regular exercise, diabetes mellitus, and history of taking medication for dyslipidemia within a year. Incidence rate was described by per 100,000 PYs. Significant values in Q4 were marked bold. TC, total cholesterol; HDL-C, high-density lipoprotein cholesterol; LDL-C, low-density lipoprotein cholesterol; IR, incidence rate; PY, person years; aHR, adjusted hazard ratio; CI, confidence interval; TG, triglycerides.

**Table 3 biomedicines-10-01617-t003:** Risk of subtypes of hematologic malignancy by quartile of each lipid profile.

		Multiple Myeloma (*n* = 1654)		Hodgkin’s Lymphoma (*n* = 189)		Non-Hodgkin’s Lymphoma (*n* = 1965)		Lymphoid Leukemia (*n* = 437)		Myeloid Leukemia (*n* = 1501)	
		Pre	Post	Pre	Post	Pre	Post	Pre	Post	Pre	Post
TC	Q1	1 (Ref.)	1 (Ref.)	1 (Ref.)	1 (Ref.)	1 (Ref.)	1 (Ref.)	1 (Ref.)	1 (Ref.)	1 (Ref.)	1 (Ref.)
	Q2	0.88 (0.64–1.21)	0.72 (0.62–0.84)	1.08 (0.44–2.67)	0.61 (0.38–0.97)	0.95 (0.75–1.21)	0.85 (0.73–0.98)	0.81 (0.52–1.27)	1.00 (0.73–1.38)	0.78 (0.60–1.01)	0.84 (0.71–1.00)
	Q3	0.77 (0.54–1.09)	0.67 (0.58–0.78)	2.38 (1.07–5.31)	0.71 (0.46–1.10)	0.93 (0.72–1.20)	0.85 (0.74–0.98)	0.62 (0.37–1.04)	0.83 (0.60–1.15)	0.98 (0.76–1.26)	0.74 (0.62–0.88)
	Q4	0.66 (0.44–0.99)	0.62 (0.54–0.72)	1.30 (0.46–3.66)	0.52 (0.33–0.82)	1.02 (0.78–1.35)	0.69 (0.59–0.79)	0.68 (0.39–1.19)	0.73 (0.53–1.01)	0.86 (0.64–1.15)	0.70 (0.59–0.83)
HDL-C	Q1	1 (Ref.)	1 (Ref.)	1 (Ref.)	1 (Ref.)	1 (Ref.)	1 (Ref.)	1 (Ref.)	1 (Ref.)	1 (Ref.)	1 (Ref.)
	Q2	0.69 (0.47–1.00)	0.83 (0.72–0.95)	1.00 (0.36–2.76)	1.03 (0.69–1.54)	1.05 (0.81–1.37)	0.90 (0.78–1.03)	1.16 (0.70–1.92)	0.82 (0.61–1.09)	0.92 (0.69–1.21)	0.84 (0.71–0.98)
	Q3	0.86 (0.61–1.22)	0.75 (0.65–0.86)	1.36 (0.54–3.44)	0.96 (0.63–1.46)	0.93 (0.71–1.21)	0.93 (0.82–1.07)	0.85 (0.50–1.45)	0.86 (0.64–1.15)	1.01 (0.77–1.32)	0.86 (0.73–1.01)
	Q4	0.62 (0.47–0.99)	0.57 (0.49–0.67)	1.36 (0.53–3.48)	0.46 (0.26–0.81)	0.80 (0.61–1.06)	0.83 (0.71–0.96)	0.85 (0.49–1.45)	0.74 (0.53–1.02)	0.80 (0.60–1.07)	0.80 (0.67–0.95)
LDL-C	Q1	1 (Ref.)	1 (Ref.)	1 (Ref.)	1 (Ref.)	1 (Ref.)	1 (Ref.)	1 (Ref.)	1 (Ref.)	1 (Ref.)	1 (Ref.)
	Q2	0.78 (0.56–1.10)	0.80 (0.69–0.94)	1.19 (0.52–2.71)	0.73 (0.45–1.16)	0.92 (0.72–1.18)	0.93 (0.80–1.07)	1.28 (0.81–2.02)	0.89 (0.64–1.23)	0.95 (0.73–1.22)	0.92 (0.77–1.09)
	Q3	0.91 (0.66–1.27)	0.83 (0.72–0.97)	1.52 (0.67–3.42)	0.67 (0.42–1.06)	0.97 (0.75–1.25)	0.86 (0.75–1.00)	0.90 (0.54–1.51)	0.90 (0.66–1.24)	1.00 (0.77–1.30)	0.85 (0.71–1.01)
	Q4	0.62 (0.41–0.93)	0.71 (0.62–0.83)	0.85 (0.29–2.50)	0.66 (0.43–1.03)	1.06 (0.80–1.39)	0.73 (0.63–0.84)	0.85 (0.47–1.52)	0.75 (0.54–1.02)	0.94 (0.70–1.26)	0.77 (0.65–0.91)
TG	Q1	1 (Ref.)	1 (Ref.)	1 (Ref.)	1 (Ref.)	1 (Ref.)	1 (Ref.)	1 (Ref.)	1 (Ref.)	1 (Ref.)	1 (Ref.)
	Q2	1.15 (0.83–1.60)	1.02 (0.86–1.22)	0.82 (0.36–1.89)	1.03 (0.59–1.79)	1.19 (0.94–1.51)	0.98 (0.82–1.16)	1.01 (0.63–1.61)	1.40 (0.96–2.04)	1.10 (0.85–1.42)	0.94 (0.77–1.14)
	Q3	0.83 (0.57–1.22)	1.03 (0.87–1.22)	1.09 (0.47–2.53)	0.95 (0.56–1.62)	1.02 (0.78–1.33)	1.10 (0.94–1.30)	0.92 (0.55–1.54)	1.28 (0.89–1.86)	0.97 (0.73–1.28)	0.92 (0.76–1.11)
	Q4	1.14 (0.78–1.68)	1.03 (0.87–1.22)	1.36 (0.55–3.39)	1.06 (0.63–1.78)	1.14 (0.85–1.52)	0.96 (0.82–1.13)	1.04 (0.60–1.81)	1.05 (0.72–1.52)	1.32 (0.99–1.76)	0.89 (0.74–1.08)

Adjusted hazard ratio and 95% confidence interval was shown in Table 3. Hazard ratio was adjusted for age, body mass index, smoking, alcohol consumption, regular exercise, diabetes mellitus, and history of taking medication for dyslipidemia within a year. Significant values in Q4 were marked bold. TC, Total cholesterol; HDL-C, high-density lipoprotein cholesterol; LDL-C, low-density lipoprotein cholesterol; TG, TG; Pre, premenopausal women; Post, postmenopausal women.

**Table 4 biomedicines-10-01617-t004:** Risk of hematologic malignancy by quartile of each lipid profile according to statin use and menopausal status.

			Total (*n* = 2,811,410)				Premenopause (*n* = 1,189,806)				Postmenopause (*n* = 1,621,604)			
Statin Use			Subjects (N)	Event (*n)*	IR	Model 3 aHR (95% CI)	Subjects (N)	Event (*n*)	IR	Model 3 aHR (95% CI)	Subjects (N)	Event (*n*)	IR	Model 3 aHR (95% CI)
TC	No	Q1	612,024	1152	227.8	1 (Ref.)	390,151	388	119.6	1 (Ref.)	221,873	764	421.2	1 (Ref.)
		Q2	633,493	1133	216.0	0.82 (0.76–0.89)	327,016	298	109.6	0.84 (0.72–0.98)	306,477	835	330.6	0.78 (0.71–0.86)
		Q3	648,396	1234	229.8	0.79 (0.73–0.85)	263,987	281	128.0	0.92 (0.78–1.07)	384,409	953	300.2	0.71 (0.65–0.78)
		Q4	634,419	1187	226.2	0.69 (0.64–0.75)	178,483	195	131.5	0.85 (0.72–1.02)	455,936	992	263.5	0.62 (0.56–0.68)
	Yes	Q1	98,505	290	360.0	1 (Ref.)	9630	8	100.0	1 (Ref.)	88,875	282	388.7	1 (Ref.)
		Q2	59,476	144	294.6	0.85 (0.69–1.04)	5928	6	121.7	1.15 (0.40–3.33)	53,548	138	314.0	0.84 (0.68–1.03)
		Q3	51,631	133	313.0	0.93 (0.76–1.14)	5827	6	123.5	1.14 (0.40–3.31)	45,804	127	337.4	0.92 (0.75–1.14)
		Q4	73,466	176	290.6	0.90 (0.74–1.09)	8784	6	82.0	0.74 (0.25–2.14)	64,682	170	319.3	0.90 (0.74–1.09)
P interaction			0.057				0.863				0.003			
HDL-C	No	Q1	630,164	1551	298.3	1 (Ref.)	233,843	279	143.2	1 (Ref.)	396,321	1272	391.1	1 (Ref.)
		Q2	608,829	1155	229.1	0.86 (0.80–0.93)	270,038	272	121.1	0.88 (0.75–1.05)	338,791	883	316.0	0.86 (0.79–0.94)
		Q3	663,518	1169	212.7	0.86 (0.80–0.93)	325,834	335	123.8	0.93 (0.79–1.09)	337,684	834	299.1	0.85 (0.77–0.92)
		Q4	625,821	831	160.4	0.70 (0.64–0.76)	329,922	276	100.8	0.78 (0.66–0.93)	295,899	555	227.1	0.67 (0.61–0.74)
	Yes	Q1	81,964	247	368.6	1 (Ref.)	7907	2	30.4	1 (Ref.)	74,057	245	405.4	1 (Ref.)
		Q2	71,019	177	303.0	0.85 (0.70–1.04)	7312	10	164.3	5.37 (1.17–24.57)	63,707	167	319.2	0.81 (0.67–0.99)
		Q3	70,148	171	296.2	0.85 (0.70–1.04)	7617	7	110.3	3.58 (0.74–17.35)	62,531	164	319.1	0.83 (0.68–1.01)
		Q4	59,947	148	300.9	0.89 (0.73–1.10)	7333	7	114.7	3.59 (0.73–17.70)	52,614	141	326.1	0.87 (0.71–1.07)
P interaction			0.084				0.129				0.059			
LDL-C	No	Q1	589.099	1.058	217.5	1 (Ref.)	362,532	349	115.9	1 (Ref.)	226,567	709	382.6	1 (Ref.)
		Q2	630.644	1.120	214.6	0.89 (0.82–0.97)	331,150	315	114.4	0.92 (0.79–1.07)	299,494	805	326.5	0.86 (0.78–0.95)
		Q3	664.971	1.308	237.5	0.89 (0.82–0.96)	279,213	295	127.1	0.95 (0.81–1.11)	385,758	1013	318.1	0.83 (0.76–0.92)
		Q4	643.618	1.220	229.0	0.76 (0.70–0.83)	186,742	203	130.7	0.89 (0.75–1.06)	456,876	1017	269.4	0.70 (0.64–0.77)
	Yes	Q1	112,423	322	350.2	1 (Ref.)	11,093	10	108.6	1 (Ref.)	101,330	312	377.1	1 (Ref.)
		Q2	58,510	151	314.0	0.93 (0.76–1.13)	5890	5	102.0	0.91 (0.31–2.68)	52,620	146	338.0	0.93 (0.76–1.13)
		Q3	47,236	117	300.8	0.92 (0.74–1.14)	5364	5	111.8	0.97 (0.33–2.86)	41,872	112	325.4	0.92 (0.74–1.14)
		Q4	64,909	153	285.6	0.91 (0.75–1.10)	7822	6	91.9	0.80 (0.29–2.21)	57,087	147	312.5	0.91 (0.75–1.11)
P interaction			0.426				0.995				0.141			
TG	No	Q1	680,201	911	161.3	1 (Ref.)	433,299	372	103.3	1 (Ref.)	246,902	539	263.5	1 (Ref.)
		Q2	630,419	1112	213.0	1.08 (0.99–1.18)	316,092	326	124.1	1.11 (0.96–1.29)	314,327	786	303.1	1.05 (0.94–1.17)
		Q3	621,451	1312	255.5	1.11 (1.02–1.21)	239,299	236	118.6	0.99 (0.84–1.17)	382,152	1,076	342.1	1.11 (1.00–1.23)
		Q4	596,261	1371	278.9	1.07 (0.98–1.16)	170,947	228	160.5	1.23 (1.03–1.46)	425,314	1,143	327.0	1.01 (0.91–1.12)
	Yes	Q1	33,358	103	374.4	1 (Ref.)	4854	6	148.6	1 (Ref.)	28,504	97	413.2	1 (Ref.)
		Q2	56,950	130	277.4	0.70 (0.54–0.90)	6325	2	37.9	0.24 (0.05–1.21)	50,625	128	307.7	0.72 (0.55–0.93)
		Q3	82,493	214	315.7	0.76 (0.60–0.97)	8108	9	133.3	0.84 (0.29–2.41)	74,385	205	335.9	0.76 (0.60–0.97)
		Q4	110,277	296	327.7	0.78 (0.62–0.98)	10,882	9	99.5	0.60 (0.20–1.75)	99,395	287	353.1	0.79 (0.62–1.00)
P interaction			0.004				0.194				0.013			

Model 3 was adjusted for age, body mass index, smoking, alcohol consumption, regular exercise, diabetes mellitus, and history of taking medication for dyslipidemia within a year. Incidence rate was described by per 100,000 PYs. Significant values in Q4 and P interaction were marked bold. TC, total cholesterol; HDL-C, high-density lipoprotein cholesterol; LDL-C, low-density lipoprotein cholesterol; IR, incidence rate; PY, person years; aHR, adjusted hazard ratio; CI, confidence interval; TG, triglyceride.

## Data Availability

The datasets used and/or analyzed during this study are available from the corresponding author on reasonable request.

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
