# Peer review of "Association between Cholesterol Level and the Risk of Hematologic Malignancy According to Menopausal Status: A Korean Nationwide Cohort Study"

_biomedicines, 2022, doi:10.3390/biomedicines10071617_

Round 1

Reviewer 1 Report

Jung et al. studied the association between cholesterol level and the risk of hematologic malignancy according to menopausal status with the Korean nationwide cohort study. This study finding showed that lipid profiles are differently associated with hematologic malignancy (HM) risk by menopausal status. Further, this study shows that overall, 0.19% developed HM. Postmenopausal women showed inverse association with total cholesterol, HDL-c, LDL-C and the risk of overall HM. In the case of premenopausal women highest quartile, HDL-c was associated with a reduced risk of HM, but the lowest quartile of HDL-c was consistent with postmenopausal women's results. Triglyceride increased the risk of HM compared to the lowest quartile of TG, only in premenopausal women.

This was an interesting finding since the literature has only limited information on how reproductive factors interact with serum cholesterol level and HM.

Comments:

1. Title may be modified (include Korean) as follow: Association between cholesterol level and the risk of hematologic malignancy according to menopausal status: A Korean nationwide cohort study

2. Measurement of Cholesterol levels: “The cholesterol concentrations were measured enzymatically” Does the authors sure all the clinics/hospitals followed the same method of Cholesterol estimation for the entire included participant? If yes, include the method reference. If no, include the limitation.

3. Page 11: Discussion: “, low levels of total cholesterol and LDL-C induced by the use of stain were not related to an increased risk of HM.”  What was that endogenous etiology behind the lipid levels takes a role in the risk of HM? But the systemic reviews state that statin has a potential preventive effect on the risk of HM, but not in this study, what was the reason? Provide some literature on this paragraph to support the finding.

Reviewer 2 Report

The manuscript by Jung et al is a very extensive association study between cholesterol levels and risk of haematological malignancy. The novelty of this study is that the authors considered the menopausal status as a principal factor in their analyses. The authors concluded that their study supported an association between increased risk of malignancies both in premenopausal and postmenopausal women. Moreover, high TG was associated with increased risk only in premenopausal women. 

As strengths, this study covers a very large sample size, complete with medical histories. Their statistical design and study is extensive, and therefore, the conclusions are well supported by their data. 

As limitations, the study does not provide insight into the mechanisms of this difference, although I believe that for a study of this type, this is not really required.

Therefore, I would recommend publication, with just minor spelling checks.
